# Model Description of Similarity-Based Recommendation Systems

**DOI:** 10.3390/e21070702

**Published:** 2019-07-17

**Authors:** Takafumi Kanamori, Naoya Osugi

**Affiliations:** 1Tokyo Institute of Technology, 2-12-1 Ookayama, Meguro-ku, Tokyo 152-8552, Japan; 2RIKEN AIP, Nihonbashi 1-chome Mitsui Building, 15th floor, 1-4-1 Nihonbashi, Chuo-ku, Tokyo 103-0027, Japan; 3Recruit Technologies Co., Ltd., GranTokyo South Tower, 1-9-2 Marunouchi, Chiyoda-ku, Tokyo 100-6640, Japan

**Keywords:** recommendation, similarity measures, bernoulli mixture models, completely positive matrix

## Abstract

The quality of online services highly depends on the accuracy of the recommendations they can provide to users. Researchers have proposed various similarity measures based on the assumption that similar people like or dislike similar items or people, in order to improve the accuracy of their services. Additionally, statistical models, such as the stochastic block models, have been used to understand network structures. In this paper, we discuss the relationship between similarity-based methods and statistical models using the Bernoulli mixture models and the expectation-maximization (EM) algorithm. The Bernoulli mixture model naturally leads to a completely positive matrix as the similarity matrix. We prove that most of the commonly used similarity measures yield completely positive matrices as the similarity matrix. Based on this relationship, we propose an algorithm to transform the similarity matrix to the Bernoulli mixture model. Such a correspondence provides a statistical interpretation to similarity-based methods. Using this algorithm, we conduct numerical experiments using synthetic data and real-world data provided from an online dating site, and report the efficiency of the recommendation system based on the Bernoulli mixture models.

## 1. Introduction

In this paper, we study recommendation problems, in particular, the *reciprocal recommendation*. The reciprocal recommendation is regarded as an edge prediction problem of random graphs. For example, a job recruiting service provides preferable matches between companies and job seekers. The corresponding graph is a bipartite graph, where nodes are categorized into two groups: job seekers and companies. Directed edges from one group to the other are the expression of the user’s interests. Using this, the job recruiting service recommends unobserved potential matches between users and companies. Another common example is online dating services. Similarly, the corresponding graph is expressed as a bipartite graph with two groups, i.e., males and females. The directed edges are the preference expressions among users. The recommendation system provides potentially preferable partners to each user. The quality of such services depends entirely on the prediction accuracy of the unobserved or newly added edges. The edge prediction has been widely studied as a class of important problems in social networks [1,2,3,4,5].

In recommendation problems, it is often assumed that similar people like or dislike similar items, people, etc. Based on this assumption, researchers have proposed various similarity measures. The similarity is basically defined through the topological structure of the graph that represents the relationship among users or items. Neighbor-based metrics, path-based metrics, and random walk based metrics are commonly used in this type of analysis. Then, a similarity matrix defined from the similarity measure is used for the recommendation. Another approach is employing the statistical models, such as stochastic block models [6], that are used to estimate network structures, such as clusters or edge distributions. The learning methods using statistical models often achieve high prediction accuracy in comparison to similarity-based methods. Details on this topic are reported in [7] and the references therein.

The main purpose of this paper is to investigate the relationship between similarity-based methods and statistical models. We show that a class of widely applied similarity-based methods can be derived from the Bernoulli mixture models. More precisely, the Bernoulli mixture model with the expectation-maximization (EM) algorithm [8] naturally derives a completely positive matrix [9] as the similarity matrix. The class of completely positive matrices is a subset of doubly nonnegative matrices, i.e.,  positive semidefinite and element-wise nonnegative matrices [10]. Additionally, we provide an interpretation of completely positive matrices as a statistical model satisfying exchangeability [11,12,13,14]. Based on the above argument, we connect the similarity measures using completely positive matrices to the statistical models. First, we prove that most of the commonly used similarity measures yield completely positive matrices as the similarity matrix. Then, we propose an algorithm that transforms the similarity matrix to the Bernoulli mixture model. As a result, we obtain a statistical interpretation of the similarity-based methods through the Bernoulli mixture models. We conduct numerical experiments using synthetic data and real-world data provided from an online dating site, and  report the efficiency of the recommendation method based on the Bernoulli mixture models.

Throughout the paper, the following notation is used. Let [n] be {1,…,n} for a positive integer *n*. For the matrices *A* and *B*, A≤B denotes the element-wise inequality and 0≤A denotes that *A* is entry-wise non-negative. The same notation is used for the comparison of vectors. The Euclidean norm (resp. 1-norm) is denoted as ∥a∥ (resp. ${∥a∥}_{1}$). For the symmetric matrix *A*, O⪯A means that *A* is positive semidefinite.

In this paper, we mainly focus on directed bipartite graphs. The directed bipartite graph G=(X,Y,E) consists of the disjoint sets of nodes, $X=\left\{x_{1},\ldots ,x_{n}\right\},Y=\left\{y_{1},\ldots ,y_{m}\right\}$, and the set of directed edges E⊂(X×Y)∪(Y×X). The sizes of *X* and *Y* are *n* and *m*, respectively. Using the matrices $A=\left(a_{ij}\right)\in {\left\{0,1\right\}}^{n\times m}$ and $B=\left(b_{ji}\right)\in {\left\{0,1\right\}}^{m\times n}$, the adjacency matrix of *G* is given by
$$\begin{array}{c} \tilde{A}=\left(\begin{array}{cc} O & A\\ B & O \end{array}\right)\in {\left\{0,1\right\}}^{\left(n+m)\times (n+m\right)}, \end{array}$$
where $a_{ij}=1$ (respectively $b_{ji}=1$) if and only if $(x_{i},y_{j})\in E$ (respectively $(y_{j},x_{i})\in E$). For the directed graph, the adjacency matrix $\tilde{A}$ is not necessarily symmetric. In many social networks, each node of the graph corresponds to each user with attributes such as the age, gender, preferences, etc. In this paper, an observed attribute associated with the node $x_{i}\in X$ (resp. $y_{j}\in Y$) is expressed by a multi-dimensional vector $x_{i}$ (resp. $y_{j}$). In real-world networks, the attributes are expected to be closely related to the graph structure.

## 2. Recommendation with Similarity Measures

We introduce similarity measures commonly used in the recommendation. Let us consider the situation that each element in *X* sends messages to some elements in *Y*, and vice versa. The messages are expressed as directed edges between *X* and *Y*. The observation is, thus, given as a directed bipartite graph G=(X,Y,E). The directed edge between nodes is called the expression of interest (EOI) in the context of the recommendation problems [15]. The purpose is to predict an unseen pair (x,y)∈X×Y such that these two nodes will send messages to each other. This problem is called the reciprocal recommendation [15,16,17,18,19,20,21,22,23,24,25]. In general, the graph is sparse, i.e., the number of observed edges is much fewer than the number of all possible edges.

In social networks, similar people tend to like and dislike similar people, and are liked and disliked by similar people as studied in [15,26]. Such observations motivated us to define similarity measures. Let $\mathrm{sim}(i,i^{\prime })$ be a similarity measure between the nodes $x_{i},x_{i^{\prime }}\in X$. In a slight abuse of notation, we write $\mathrm{sim}(j,j^{\prime })$ to indicate a similarity measure between the nodes $y_{j},y_{j^{\prime }}\in Y$. Based on the observed EOIs, the score of $x_{i}$’s interest to $y_{j}\in Y$ for i∈[n] is defined as
(1)$$\begin{array}{c} \mathrm{score}\left(i\to j\right)=\frac{1}{n}\sum_{i^{\prime }\in \left[n\right]}\mathrm{sim}\left(i,i^{\prime }\right)a_{i^{\prime }j}. \end{array}$$
If $x_{i^{\prime }}\in X$ is similar to $x_{i}$ and the edge $\left(x_{i^{\prime }},y_{j}\right)$ exists, the user $x_{i}$ gets a high score even if $(x_{i},y_{j})\notin E$. In the reciprocal recommendation, score(j→i) defined by
(2)$$\begin{array}{c} \mathrm{score}\left(j\to i\right)=\frac{1}{m}\sum_{j^{\prime }\in \left[m\right]}\mathrm{sim}\left(j,j^{\prime }\right)b_{j^{\prime }i} \end{array}$$
is also important. The reciprocal score between $x_{i}$ and $y_{j}$, score(i∼j), is defined as the harmonic mean of score(i→j) and score(j→i) [15]. This is employed to measure the affinity between $x_{i}$ and $y_{j}$.

Table 1 shows popular similarity measures including graph-based measures and a content-based measure [1]. For the node $x_{i}\in X$ in the graph G=(X,Y,E), let $s_{i}$ (resp. ${\bar{s}}_{i}$) be the index set of outer-edges, $\left\{j|\left(x_{i},y_{j}\right)\in E\right\}$ (resp. in-edges, $\left\{j|\left(y_{j},x_{i}\right)\in E\right\}$) and |s| be the cardinality of the finite set *s*. In the following, the similarity measures based on outer-edges are introduced on directed bipartite graphs. The set of outer-edges $s_{i}$ can be replaced with ${\bar{s}}_{i}$ to define the similarity measure based on in-edges.

In graph-based measures, the similarity between the nodes $x_{i}$ and $x_{i^{\prime }}$ is defined based on $s_{i}$ and $s_{i^{\prime }}$. Some similarity measures depend only on $s_{i}$ and $s_{i^{\prime }}$, and others may depend on the whole topological structure of the graph. In Table 1, the first group includes the Common Neighbors, Parameter-Dependent, Jaccard Coefficient, Sørensen Index, Hub Depressed, and Hub Promoted. The similarity measures in this group are locally defined, i.e., $\mathrm{sim}(i,i^{\prime })$ depends only on $s_{i}$ and $s_{i^{\prime }}$. The second group includes SimRank, Adamic-Adar coefficient, and Resource Allocation. They are also defined from the graph structure. However, the similarity between the nodes $x_{i}$ and $x_{i^{\prime }}$ depends on the topological structure more than $s_{i}$ and $s_{i^{\prime }}$. The third group consists of the content-based similarity, which is defined by the attributes associated with each node.

Below, we supplement the definition of the SimRank and the content-based similarity.

SimRank:

SimRank [33] and its reduced variant [35] are determined from the random walk on the graph. Hence, the similarity between the two nodes depends on the whole structure of the graph. For c∈(0,1), the similarity matrix $\tilde{S}={\left({\tilde{S}}_{ij}\right)}_{i,j\in [n+m]}$ on X∪Y is given as the solution of
$$\begin{array}{c} {\tilde{S}}_{ii^{\prime }}=c\sum_{k\in s_{i},k^{\prime }\in s_{i^{\prime }}}\frac{{\tilde{S}}_{kk^{\prime }}}{|s_{i}\left|\right|s_{i^{\prime }}|} \end{array}$$
for $i\neq i^{\prime }$, while the diagonal element ${\tilde{S}}_{ii}$ is fixed to 1. Let $P\in {\left[0,1\right]}^{(n+m)\times n+m}$ be the column-normalized adjacency matrix defined from the adjacency matrix of G=(X,Y,E). Then, $\tilde{S}$ satisfies $\tilde{S}=cP^{T}\tilde{S}P+D$, where *D* is a diagonal matrix satisfying $1-c\leq D_{ii}\leq 1$. In the reduced SimRank, *D* is defined as (1−c)I. For the bipartite graph, the similarity matrix based on the SimRank is given as a block diagonal matrix.

Content-Based Similarity:

In RECON [17,21], the content-based similarity measure is employed. Suppose that $x_{i}={\left(x_{ia}\right)}_{a\in [Q]}\in \prod_{a\in [Q]}V_{a}$ is the attributes of the node $x_{i}$, where $V_{a},a\in \left[Q\right]$ are finite sets and $x_{ia}\in V_{a}$. The continuous variables in the features are appropriately discredited. The similarity measure is defined using the number of shared attributes, i.e., 
$$\begin{array}{c} \mathrm{sim}\left(i,i^{\prime }\right)=\frac{1}{Q}\sum_{a\in [Q]}1\left[x_{ia}=x_{i^{\prime }a}\right]=\frac{1}{Q}\sum_{a\in [Q]}\sum_{b\in V_{a}}1\left[x_{ia}=b\right]\cdot 1\left[x_{i^{\prime }a}=b\right]. \end{array}$$
In RECON, the score is defined from the normalized similarity, i.e., 
$$\begin{array}{c} \mathrm{score}\left(i\to j\right)=\sum_{j^{\prime }}\frac{a_{ij^{\prime }}}{\sum_{k}a_{ik}}\mathrm{sim}\left(j^{\prime },j\right). \end{array}$$

The similarity-based recommendation is simple but the theoretical properties have not been sufficiently studied. In the next section, we introduce statistical models and consider the relationship to similarity-based methods.

## 3. Bernoulli Mixture Models and Similarity-Based Prediction

In this section, we show that the similarity-based methods are derived from Bernoulli mixture models (BMMs). BMMs have been employed in some studies [36,37,38] for block clustering problems, Here, we show that the BMMs are also useful for recommendation problems.

Suppose that each node belongs to a class c∈[C]. Let $\pi_{c}$ (respectively $\pi_{c}^{\prime }$) be the probability that each node in *X* (respectively *Y*) belongs to the class c∈[C]. We assume that the class at each node is independently drawn from these probability distributions. Though the number of classes, *C*, can be different in each group, here we suppose that they are the same for simplicity. When the node $x_{i}\in X$ in the graph belongs to the class *c*, the occurrence probability of the directed edge from $x_{i}$ to $y_{j}\in Y$ is defined by the Bernoulli distribution with the parameter $\alpha_{cj}\in \left(0,1\right)$. As previously mentioned, the adjacency matrix of the graph consists of $A=(a_{ij})$ and $B=(b_{ji})$. We assume that all elements of *A* and *B* are independently distributed. For each $x_{i}\in X$, the probability of ${\left(a_{ij}\right)}_{j\in [m]}$ is given by the BMM,
$$\begin{array}{c} \sum_{c\in [C]}\pi_{c}\prod_{j\in [m]}\alpha_{cj}^{a_{ij}}{\left(1-\alpha_{cj}\right)}^{1-a_{ij}}, \end{array}$$
and the probability of the adjacency submatrix $A=(a_{ij})$ is given by
(3)$$\begin{array}{c} P\left(A\right)=\prod_{i\in [n]}\sum_{c\in [C]}\pi_{c}\prod_{j\in [m]}\alpha_{cj}^{a_{ij}}{\left(1-\alpha_{cj}\right)}^{1-a_{ij}}. \end{array}$$

In the same way, the probability of the adjacency submatrix *B* is given by
(4)$$\begin{array}{c} P\left(B\right)=\prod_{j\in [m]}\sum_{c\in [C]}\pi_{c}^{\prime }\prod_{i\in [n]}\beta_{ci}^{b_{ji}}{\left(1-\beta_{ci}\right)}^{1-b_{ji}}, \end{array}$$
where $\beta_{ci}\in \left(0,1\right)$ is the parameter of the Bernoulli distribution. Hence, the probability of the whole adjacency matrix $\tilde{A}$ is given by
(5)$$\begin{array}{c} P\left(\tilde{A};\Psi \right)=P\left(A\right)P\left(B\right), \end{array}$$
where Ψ is the set of all parameters in the BMM, i.e.,  $\pi_{c},\pi_{c}^{\prime },\alpha_{cj}$ and $\beta_{ci}$ for i∈[n],j∈[m] and c∈[C]. One can introduce the prior distribution on the parameter $\alpha_{cj}$ and $\beta_{cj}$. The beta distribution is commonly used as the conjugate prior to the Bernoulli distribution.

The parameter is estimated by maximizing the likelihood for the observed adjacency matrix $\tilde{A}$. The probability $P(\tilde{A};\Psi )$ is decomposed into two probabilities, P(A) and P(B), which do not share the parameters. In fact, P(A) depends only on $\pi_{c}$ and $\alpha_{cj}$ and P(B) depends only on $\pi_{c}^{\prime }$ and $\beta_{cj}$. In the following, we consider the parameter estimation of P(A). The same procedure works for the estimation of the parameters in P(B).

The expectation-maximization (EM) algorithm [8] can be used to calculate the maximum likelihood estimator. The auxiliary variables used in the EM algorithm have an important role in connecting the BMM with the similarity-based recommendation methods. Using the Jensen’s inequality, we find that the log-likelihood logP(A) is bounded below as
(6)$$\begin{array}{ll} \log P(A) & =\sum_{i}\log\sum_{c}r\left(c|i\right)\frac{\pi_{c}\prod_{j\in Y}\alpha_{cj}^{a_{ij}}{\left(1-\alpha_{cj}\right)}^{1-a_{ij}}}{r(c|i)}\\ & \geq J\left(r,\Psi ;A\right):=\sum_{i,c}r\left(c|i\right)\log\frac{\pi_{c}}{r(c|i)}+\sum_{i,j,c}r\left(c|i\right)\log\alpha_{cj}^{a_{ij}}{\left(1-\alpha_{cj}\right)}^{1-a_{ij}}, \end{array}$$
where the parameter $r={\left(r\left(c|i\right)\right)}_{c,i}$ is positive auxiliary variables satisfying $\sum_{c\in [C]}r\left(c|i\right)=1$. In the above inequality, the equality holds when r(c|i) is proportional to $\pi_{c}\prod_{j}\alpha_{cj}^{a_{ij}}{\left(1-\alpha_{cj}\right)}^{1-a_{ij}}$. The auxiliary variable r(c|i) is regarded as the class probability of $x_{i}\in X$ when the adjacency matrix is observed.

In the EM algorithm, the lower bound of the log-likelihood, i.e., the function J(r,Ψ;A) in (Equation 6) is maximized. For this purpose, the alternating optimization method is used. Firstly the parameter Ψ is optimized for the fixed *r*, and secondly, the parameter *r* is optimized for the fixed Ψ. This process is repeatedly conducted until the function value J(r,Ψ;A) converges. Importantly, in each iteration the optimal solution is explicitly obtained. The following is the learning algorithm of the parameters:(7)$$\begin{array}{cc} \mathrm{Step}\;1: & \;\left\{\begin{array}{cc} \pi_{c}\; & ⟵\;\frac{1}{n}\sum_{i}r\left(c|i\right),\\ \alpha_{cj}\; & ⟵\;\frac{1}{n\pi_{c}}\sum_{i}r\left(c|i\right)a_{ij}, \end{array}\right. \end{array}$$
(8)$$\begin{array}{cc} \mathrm{Step}\;2: & \;r\left(c|i\right)\;⟵\;\frac{\pi_{c}\prod_{j}\alpha_{cj}^{a_{ij}}{\left(1-\alpha_{cj}\right)}^{1-a_{ij}}}{\sum_{c}\pi_{c}\prod_{j}\alpha_{cj}^{a_{ij}}{\left(1-\alpha_{cj}\right)}^{1-a_{ij}}}. \end{array}$$

The estimator of the parameter Ψ is obtained by repeating (Equation 7) and (Equation 8).

Using the auxiliary variables r(c|i), one can naturally define the “occurrence probability” of the edge from $x_{i}$ to $y_{j}$. Here, the occurrence probability is denoted by score(i→j). Note that the auxiliary variable r(c|i) is regarded as the conditional probability that $x_{i}$ belongs to the class *c*. If $x_{i}$ belongs to the class *c*, the occurrence probability of the edge $\left(x_{i},y_{j}\right)$ is $\alpha_{cj}$. Hence, the occurrence probability of the edge $\left(x_{i},y_{j}\right)$ is naturally given by
(9)$$\begin{array}{ll} \mathrm{score}(i\to j) & :=\sum_{c}r\left(c|i\right)\alpha_{cj}=\sum_{c}r\left(c|i\right)\frac{1}{n\pi_{c}}\sum_{i^{\prime }}r\left(c|i^{\prime }\right)a_{i^{\prime }j}\\ & =\frac{1}{n}\sum_{i^{\prime }}\mathrm{sim}\left(i,i^{\prime }\right)a_{i^{\prime }j}, \end{array}$$
where the updated parameter $\alpha_{cj}$ in (Equation 7) is substituted. The similarity measure $\mathrm{sim}(i,i^{\prime })$ on *X* in the above is defined by (10)$$\begin{array}{c} \mathrm{sim}\left(i,i^{\prime }\right)=\sum_{c}\frac{r\left(c|i\right)r\left(c|i^{\prime }\right)}{\pi_{c}}=\frac{\sum_{c}\pi_{c}r\left(i|c\right)r\left(i^{\prime }|c\right)}{r\left(i\right)r\left(i^{\prime }\right)}=\frac{r(i,i^{\prime })}{r\left(i\right)r\left(i^{\prime }\right)}, \end{array}$$
where $r\left(i\right):=1/n,r\left(i|c\right):=r\left(c|i\right)r\left(i\right)/\pi_{c}$ and
(11)$$\begin{array}{c} r\left(i,i^{\prime }\right):=\sum_{c\in [C]}\pi_{c}r\left(i|c\right)r\left(i^{\prime }|c\right). \end{array}$$

The equality $\sum_{i}r\left(i,i^{\prime }\right)=r\left(i^{\prime }\right)=1/n$ holds for *r* satisfying the update rule (Equation 7). The above joint probability $r(i,i^{\prime })$ clearly satisfies the symmetry, $r\left(i,i^{\prime }\right)=r\left(i^{\prime },i\right)$. This property is the special case of the finite exchangeability [11,13]. The exchangeability is related to the de Finetti’s theorem [39], and  the statistical models with the exchangeability have been used in several problems such as Bayes modeling and classification [12,40,41]. Here, we use the finite exchangeable model for the recommendation systems.

Equation (9) gives an interpretation of the heuristic recommendation methods (Equation 1) using similarity measures. Suppose that a similarity measure $\mathrm{sim}(i,i^{\prime })$ is used for the recommendation. Let us assume that the corresponding similarity matrix $S={\left(\mathrm{sim}\left(i,i^{\prime }\right)\right)}_{i,i^{\prime }\in \left[n\right]}$ is approximately decomposed into the form of the mixture model *r* in (10), i.e.,
(12)$$\begin{array}{c} S_{ii^{\prime }}\approx \frac{\sum_{c\in [C]}\pi_{c}r\left(i|c\right)r\left(i^{\prime }|c\right)}{r\left(i\right)r\left(i^{\prime }\right)}. \end{array}$$

Then, score(i→j) defined from *S* is approximately the same as that computed from the Bernoulli mixture model with the parameter Ψ that maximizes J(r,Ψ;A) for the fixed r(c|i) associated with *S*. On the other hand, the score for the recommendation computed from the Bernoulli mixture uses the maximum likelihood estimator Ψ that attains the maximum value of J(r,Ψ;A) under the optimal auxiliary parameter r(c|i). Hence, we expect that the learning method using the Bernoulli mixture model will achieve higher prediction accuracy as compared to the similarity-based methods, if the Bernoulli mixture model approximates the underling probability distribution of the observed data.

For $i,i^{\prime }\in \left[n\right]$, the probability function $r(i,i^{\prime })$ satisfying (11) leads to the n×n positive semidefinite matrix ${\left(r\left(i,i^{\prime }\right)\right)}_{i,i^{\prime }\in \left[n\right]}$ with nonnegative elements. As a result, the ratio $r\left(i,i^{\prime }\right)/r\left(i\right)r\left(i^{\prime }\right)$ is also positive semidefinite with nonnegative elements. Let us consider whether the similarity measures in Table 1 yield the similarity matrix with expression (10). Next, we demonstrate that the commonly used similarity measures meet the assumption (12) under a minor modification.

## 4. Completely Positive Similarity Kernels

For the set of all *n* by *n* symmetric matrices $S_{n}$, let us introduce two subsets of $S_{n}$; one is the completely positive matrices and the other is doubly nonnegative matrices. The set of completely positive matrices is defined as $C_{n}=\left\{S\in S_{n}|\exists N\geq 0\;s.t.\;S=NN^{T}\right\}$, and the set of doubly nonnegative matrices is defined as $D_{n}=\left\{S\in S_{n}|S\geq 0,\;S⪰O\right\}$. Though the number of columns of the matrix *N* in the completely positive matrix is not specified, it can be bounded above by n(n+1)/2+1. This is because $C_{n}$ is expressed as the convex hull of the set of rank one matrices $\left\{qq^{T}|q\in R^{n},\;q\geq 0\right\}$ as show in [11]. The Carathéodory’s theorem can be applied to prove the assertion. More detailed analysis of the matrix rank for the completely positive matrices is provided by [42]. Clearly, the completely positive matrix is doubly nonnegative matrix. However, [10] proved that there is a gap between the doubly nonnegative matrix and completely positive matrix when n≥5.

The similarity measure that yields the doubly nonnegative matrix satisfies the definition of the kernel function [43]. The kernel function is widely applied in machine learning and statistics [43]. Here, we define the completely positive similarity kernel (CPSK) as the similarity measure that leads to the completely positive matrix as the Gram matrix or similarity matrix. We consider whether the similarity measures in Table 1 yield the completely positive matrix. For such similarity measures, the relationship to the BMMs is established via (10).

**Lemma** **1.**
*(i) Let $B=(b_{ij})$ and $C=(c_{ij})$ be n×n completely positive matrices. Then, their Hadamard product $B\circ C={\left(b_{ij}c_{ij}\right)}_{i,j\in [n]}$ is completely positive. (ii) Let $\left\{B_{k}\right\}\subset C_{n}$ be a sequence of n×n completely positive matrices and define $B=\lim_{k\to \infty }B_{k}$. Then, B is the completely positive matrix.*


**Proof of Lemma** **1.**(i) Suppose that $B=FF^{T}$ and $C=GG^{T}$ such that $F=\left(f_{ik}\right)\in R^{n\times p}$ and $G=\left(g_{i\ell }\right)\in R^{n\times q}$. Then, ${\left(B\circ C\right)}_{ij}=\sum_{k}f_{ik}f_{jk}\sum_{\ell }g_{i\ell }g_{j\ell }=\sum_{k,\ell }\left(f_{ik}g_{i\ell }\right)\left(f_{jk}g_{j\ell }\right)$. Hence, the matrix $H=\left(h_{i,j}\right)\in R^{n\times pq}$ such that $h_{i,(k-1)q+\ell }=f_{ik}g_{i\ell }\geq 0$ satisfies $B\circ C=HH^{T}$. (ii) It is clear that $C_{n}$ is a closed set. ☐

Clearly, the linear sum of the completely positive matrices with non-negative coefficients yields completely positive matrices. Using this fact with the above lemma, we show that all measures in Table 1 except the HP measure are the CPSK. In the following, let $a_{i}={\left(a_{i1},\ldots ,a_{im}\right)}^{T}\in {\left\{0,1\right\}}^{m}$ for i∈[n] be non-zero binary column vectors, and let *A* be the matrix $A=\left(a_{ij}\right)\in {\left\{0,1\right\}}^{n\times m}$. The index set $s_{i}$ is defined as $s_{i}=\left\{k\;|\;a_{ik}=1\right\}\subset \left[m\right]$.

Common Neighbors

The elements of the similarity matrix are given by
$$\begin{array}{c} S_{ii^{\prime }}=\left|s_{i}\cap s_{i^{\prime }}\right|=a_{i}^{T}a_{i^{\prime }}\geq 0. \end{array}$$
Hence, $S=AA^{T}$ holds. The common neighbors similarity measure yields the CPSK.

Parameter-Dependent:

The elements of the similarity matrix are given by
$$\begin{array}{c} S_{ii^{\prime }}=\frac{a_{i}^{T}a_{i^{\prime }}}{∥a_{i}{∥}_{1}^{\lambda }{∥a_{i^{\prime }}∥}_{1}^{\lambda }}. \end{array}$$

Hence, we have $S=DAA^{T}D^{T}$, where *D* is the diagonal matrix whose diagonal elements are $1/∥a_{1}{∥}_{1}^{\lambda },\ldots ,1/{∥a_{1}∥}_{n}^{\lambda }$. The Parameter-Dependent similarity measure yields the CPSK.

Jaccard Similarity:

We have $|s_{i}\cap s_{i^{\prime }}|=a_{i}^{T}a_{i^{\prime }}$ and $|s_{i}\cup s_{i^{\prime }}|=m-{\bar{a}}_{i}^{T}{\bar{a}}_{i^{\prime }}$, where ${\bar{a}}_{i}=1-a_{i}$. Hence, the Jaccard similarity matrix $S={\left(S_{ii^{\prime }}\right)}_{i,i^{\prime }\in \left[n\right]}$ is given by
$$\begin{array}{c} S_{ii^{\prime }}=\frac{a_{i}^{T}a_{i^{\prime }}}{m-{\bar{a}}_{i}^{T}{\bar{a}}_{i^{\prime }}}. \end{array}$$

Let us define the matrices $S_{0}$ and $T_{\left(k\right)}$ respectively by ${\left(S_{0}\right)}_{ii^{\prime }}=a_{i}^{T}a_{i^{\prime }}/m$ and ${\left(T_{\left(k\right)}\right)}_{ii^{\prime }}={\left({\bar{a}}_{i}^{T}{\bar{a}}_{i^{\prime }}/m\right)}^{k}$. The matrix *S* is then expressed as $S=S_{0}\circ \sum_{k=0}^{\infty }T_{\left(k\right)}$. Lemma 1 (i) guarantees that $T_{\left(k\right)}$ is the CPSK since $T_{\left(1\right)}$ is the CPSK. Hence, the Jaccard similarity measure is the CPSK.

Sørensen Index:

The similarity matrix $S=(S_{ii^{\prime }})$ based on the Sørensen Index is given as
$$\begin{array}{c} S_{ii^{\prime }}=\frac{2a_{i}^{T}a_{i^{\prime }}}{∥a_{i}{∥}^{2}+{∥a_{i^{\prime }}∥}^{2}}=2\sum_{k=1}^{m}\int_{0}^{\infty }a_{ik}e^{-t∥a_{i}{∥}^{2}}a_{i^{\prime }k}e^{-t∥a_{i^{\prime }}{∥}^{2}}dt. \end{array}$$

The integral part is expressed as the limit of the sum of the rank one matrix, $\sum_{\ell }b_{k}\left(t_{\ell }\right)b_{k}^{T}\left(t_{\ell }^{\prime }\right)\left(t_{\ell }-t_{\ell -1}\right)$, where $t_{\ell }^{\prime }\in \left[t_{{}_{\ell }-1},t_{\ell }\right]$ and $b_{k}\left(t\right)$ is the *n*-dimensional vector defined by ${\left(b_{k}\left(t\right)\right)}_{i}=a_{ik}e^{-t∥a_{i}{∥}^{2}}\geq 0$. Hence, the Sørensen index is the CPSK.

Hub Promoted:

The hub promoted similarity measure does not yield the positive semidefinite kernel. Indeed, for the adjacency matrix
$$\begin{array}{c} A=\left(\begin{array}{ccc} 0 & 1 & 0\\ 1 & 0 & 1\\ 1 & 1 & 0 \end{array}\right) \end{array}$$
the similarity matrix based on the Hub Promoted similarity measure is given as
$$\begin{array}{c} S=\left(\begin{array}{ccc} 1 & 0 & 1\\ 0 & 1 & 1/2\\ 1 & 1/2 & 1 \end{array}\right). \end{array}$$
The eigenvalues of *S* are 1 and $(2\pm \sqrt{5})/2$. Hence, *S* is not positive semidefinite.

Hub Depressed:

The similarity matrix is defined as
$$\begin{array}{c} S_{ii^{\prime }}=\frac{a_{i}^{T}a_{i^{\prime }}}{\max\{∥a_{i}{∥}^{2},∥a_{i^{\prime }}{∥}^{2}\}}=a_{i}^{T}a_{i^{\prime }}\min\{1/∥a_{i}{∥}^{2},1/∥a_{i^{\prime }}{∥}^{2}\}. \end{array}$$
Since the min operation is expressed as the integral $\min\left\{x,y\right\}=\int_{0}^{1}1\left[t\leq x\right]\cdot 1\left[t\leq y\right]dt$ for x,y≥0, we have
$$\begin{array}{c} S_{ii^{\prime }}=\sum_{k=1}^{m}\int_{0}^{1}a_{ik}1[t\leq 1/∥a_{i}{∥}^{2}]\cdot a_{i^{\prime }k}1[t\leq 1/∥a_{i^{\prime }}{∥}^{2}]dt. \end{array}$$

In the same way as the Sørensen Index, we can prove that the Hub Depressed similarity measure is the CPSK.

SimRank:

The SimRank matrix *S* satisfies $S=cP^{T}SP+\left(1-c\right)D$ for c∈(0,1), where P≥0 is properly defined from $\tilde{A}$ and *D* is a diagonal matrix such that the diagonal element $d_{ii}$ satisfies $1-c\leq d_{ii}\leq 1$. The recursive calculation yields the equality $S=\sum_{k=0}^{\infty }c^{k}{\left(P^{k}\right)}^{T}DP^{k}$, meaning that *S* is the CPSK.

Adamic-Adar Coefficient:

Given the adjacency matrix $A=(a_{ij})$, the similarity matrix $S=(S_{ii^{\prime }})$ is expressed as
$$\begin{array}{c} S_{ii^{\prime }}=\sum_{k\in s_{i}\cap s_{i^{\prime }}}\frac{1}{\log|s_{k}^{\prime }|}=\sum_{k}\frac{a_{ik}a_{i^{\prime }k}}{\log\sum_{\ell }a_{\ell k}}, \end{array}$$
where $\frac{a_{ik}a_{i^{\prime }k}}{\log\sum_{\ell }a_{\ell k}}$ is set to zero if $\sum_{\ell }a_{\ell k}\leq 1$. Hence, we have $S=ADA^{T}$, where *D* is the diagonal matrix with the elements $D_{kk}=1/\log\sum_{\ell }a_{\ell k}$ for $\sum_{\ell }a_{\ell k}\geq 2$ and $D_{kk}=0$ otherwise. Since $S=NN^{T}$ with $N=AD^{1/2}$ holds, the similarity measure based on the Adamic-adar coefficient is the CPSK.

Resource Allocation:

In the same way as the Adamic-adar coefficient, the similarity matrix is given as
$$\begin{array}{c} S_{ii^{\prime }}=\sum_{k\in s_{i}\cap s_{i^{\prime }}}\frac{1}{|s_{k}^{\prime }|}=\sum_{k}\frac{a_{ik}a_{i^{\prime }k}}{\sum_{\ell }a_{\ell k}}, \end{array}$$
where the term $\frac{a_{ik}a_{i^{\prime }k}}{\sum_{\ell }a_{\ell k}}$ is set to zero if $\sum_{\ell }a_{\ell k}\leq 1$. We have $S=ADA^{T}$, where *D* is the diagonal matrix with the elements $D_{kk}=1/\sum_{\ell }a_{\ell k}$ for $\sum_{\ell }a_{\ell k}\geq 2$ and $D_{kk}=0$ otherwise. Since $S=NN^{T}$ with $N=AD^{1/2}$ holds, the similarity measure based on resource allocation is the CPSK.

Content-Based Similarity:

The similarity matrix is determined from the feature vector of each node as follows,
$$\begin{array}{c} S_{ii^{\prime }}=\frac{1}{Q}\sum_{a\in [Q]}\sum_{b\in V_{a}}1\left[x_{ia}=b\right]\cdot 1\left[x_{i^{\prime }a}=b\right]. \end{array}$$
Clearly, *S* is expressed as the sum of rank-one matrix $c_{a,b}c_{a,b}^{T}$, where ${\left(c_{a,b}\right)}_{i}=1\left[x_{ia}=b\right]/\sqrt{Q}\geq 0$. Hence, Content-based similarity is the CPSK.

## 5. Transformation from Similarity Matrix to Bernoulli Mixture Model

Let us consider whether the similarity matrix allows the decomposition in (10) for sufficiently large *C*. Then, we construct an algorithm providing the probability decomposition (10) that approximates the similarity matrix.

### 5.1. Decomposition of Similarity Matrix

We show that a modified similarity matrix defined from the CPSK is decomposed into the form of (10). Suppose the *n* by *n* similarity matrix *S* is expressed as (10). Then, we have
$$\begin{array}{c} S_{ii^{\prime }}r\left(i^{\prime }\right)=\frac{r(i,i^{\prime })}{r(i)}, \end{array}$$
where $r\left(i\right)=r\left(i^{\prime }\right)=1/n$. Taking the sum over $i^{\prime }\in X$, we find that the equality
(13)$$\begin{array}{c} S1/n=1 \end{array}$$
should hold. If the equality (13) is not required, the completely positive matrix *S* will be always decomposed into the form of (10) up to a constant factor. The equality (13) does not necessarily hold even when the CPSK is used. Let us define the diagonal matrix *D* as
$$\begin{array}{c} D_{ii}=\max_{i^{\prime }}{\left(S1\right)}_{i^{\prime }}-{\left(S1\right)}_{i}\geq 0,\;i\in \left[n\right], \end{array}$$
and let $\tilde{S}$ be
(14)$$\begin{array}{c} \tilde{S}=\frac{n}{\max_{i}{\left(S1\right)}_{i}}\left(S+D\right). \end{array}$$

Then, $\tilde{S}1/n=1$ holds. Since *S* is the completely positive matrix, also $\tilde{S}$ is the completely positive matrix. Suppose that $\tilde{S}/n^{2}$ is decomposed into $FF^{T}$ with the non-negative matrix $F=\left(f_{1},\ldots ,f_{C}\right)\in R^{n\times C}$. Then,
$$\begin{array}{c} \frac{1}{n^{2}}\tilde{S}=\sum_{c\in [C]}f_{c}f_{c}^{T}=\sum_{c\in [C]}{∥f_{c}∥}_{1}^{2}\frac{f_{c}}{∥f_{c}{∥}_{1}}\frac{f_{c}^{T}}{∥f_{c}{∥}_{1}}. \end{array}$$

Let us define $\pi_{c}={∥f_{c}∥}_{1}^{2}$ and $r\left(i|c\right)={\left(f_{c}\right)}_{i}/{∥f_{c}∥}_{1}\geq 0$. Since $1^{T}\tilde{S}1/n^{2}=1$, we have $\sum_{c}\pi_{c}=1$. Moreover, the equality $\tilde{S}1/n=1$ guarantees
$$\begin{array}{c} \frac{1}{n}{\left(\tilde{S}1\right)}_{i}=n\sum_{c}{∥f_{c}∥}_{1}^{2}\frac{f_{c}}{∥f_{c}{∥}_{1}}=n\sum_{c}\pi_{c}r\left(i|c\right)=1, \end{array}$$
meaning that $\sum_{c}\pi_{c}r\left(i|c\right)=r\left(i\right)=1/n$ for i∈X. Hence, we have
$$\begin{array}{c} {\tilde{S}}_{ii^{\prime }}=\frac{\sum_{c}\pi_{c}r\left(i|c\right)r\left(i^{\prime }|c\right)}{r\left(i\right)r\left(i^{\prime }\right)}. \end{array}$$
The modification (14) corresponds to the change of the balance between the self-similarity and the similarities with others.

### 5.2. Decomposition Algorithm

Let us propose a computation algorithm to obtain the approximate decomposition of the similarity matrix *S*. Once the decomposition of *S* is obtained, the recommendation using the similarity measure is connected to the BMMs. Such a correspondence provides a statistical interpretation of the similarity-based methods. For example, the conditional probability r(c|i) is available to categorize nodes into some classes according to the tendency of their preferences once a similarity matrix is obtained. The statistical interpretation provides a supplementary tool for similarity-based methods.

The problem is to find $\pi_{c}$ and r(i|c) such that the equation $\frac{\sum_{c}\pi_{c}r\left(i|c\right)r\left(i^{\prime }|c\right)}{r\left(i\right)r\left(i^{\prime }\right)}$ approximates the similarity matrix $S=S_{ii^{\prime }}$, where $r\left(i\right)=1/n=\sum_{c}\pi_{c}r\left(i|c\right)$ for n=|X|. Here, we focus on the similarity matrix on *X*. The same argument is clearly valid for the similarity matrix on *Y*.

This problem is similar to the non-negative matrix factorization (NMF) [44]. However, the standard algorithm for the NMF does not work since we have the additional constraint, $\sum_{c}\pi_{c}r\left(i|c\right)=1/n$. Here, we use the extended Kullback–Leibler (ext-KL) divergence to measure the discrepancy [45]. The ext-KL divergence between the two matrices $C=(c_{ij})$ and $D=(d_{ij})$ with nonnegative elements is defined as
(15)$$\begin{array}{c} \mathrm{KL}\left(C,D\right)=\sum_{ij}c_{ij}\log\frac{c_{ij}}{d_{ij}}-\sum_{ij}c_{ij}+\sum_{ij}d_{ij}\geq 0. \end{array}$$

The minimization of the ext-KL divergence between $S_{ii^{\prime }}$ and the model $r\left(i,i^{\prime }\right)/r\left(i\right)r\left(i^{\prime }\right)$ is formalized by
$$\begin{array}{ll} \min_{r\left(i|c\right)>0,\pi_{c}>0} & -\sum_{ij}S_{ij}\log\frac{\sum_{c}\pi_{c}r\left(i|c\right)r\left(j|c\right)}{r\left(i\right)r\left(i^{\prime }\right)}+\sum_{ij}\frac{\sum_{c}\pi_{c}r\left(i|c\right)r\left(j|c\right)}{r\left(i\right)r\left(i^{\prime }\right)},\\ & s.\;t.\;\;\sum_{c}\pi_{c}r\left(i|c\right)=r\left(i\right)=1/n,\;\;\sum_{c}\pi_{c}=1,\;\sum_{i}r\left(i|c\right)=1. \end{array}$$

This is equivalent with
(16)$$\begin{array}{ll} \min_{r\left(i|c\right)>0,\pi_{c}>0} & -\sum_{ij}S_{ij}\log\sum_{c}\pi_{c}r\left(i|c\right)r\left(j|c\right)\\ & s.\;t.\;\;\sum_{c}\pi_{c}r\left(i|c\right)=r\left(i\right)=1/n,\;\;\sum_{c}\pi_{c}=1,\;\sum_{i}r\left(i|c\right)=1. \end{array}$$

There are many optimization algorithms that can be used to solve nonlinear optimization problems with equality constants. A simple method is the alternating update of $\pi_{c}$ and r(i|c) such as the coordinate descent method [46]. Once r(i|c) is fixed, however, the parameter $\pi_{c}$ will be uniquely determined from the first equality constraint in (16) under a mild assumption. This means that the parameter $\pi_{c}$ cannot be updated while keeping the equality constants. Hence, the direct application of the coordinate descent method does not work. On the other hand, the gradient descent method with projection onto the constraint surface is a promising method [47,48]. In order to guarantee the convergence property, however, the step-length should be carefully controlled. Moreover, the projection in every iteration is computationally demanding. In the following, we propose a simple method to obtain an approximate solution of (16) with an easy implementation.

The constraint $\sum_{c}\pi_{c}r\left(i|c\right)=r\left(i\right)=1/n$ is replaced with the condition that the KL-divergence between the uniform distribution and $\sum_{c}\pi_{c}r\left(i|c\right)$ vanishes, i.e.,
$$\begin{array}{c} \frac{1}{n}\sum_{i}\log\frac{1/n}{\sum_{c}\pi_{c}r\left(i|c\right)}=0. \end{array}$$

We incorporated this constraint into the objective function to obtain tractable algorithm. Eventually, the optimization problem we considered is
$$\begin{array}{c} \max_{r\left(i|c\right)>0,\pi_{c}>0}\sum_{ij}S_{ij}\log\sum_{c}\pi_{c}r\left(i|c\right)r\left(j|c\right)+\frac{\lambda }{n}\sum_{i}\log\sum_{c}\pi_{c}r\left(i|c\right)\\ s.\;t.\;\;\;\sum_{c}\pi_{c}=1,\;\sum_{i}r\left(i|c\right)=1, \end{array}$$
where the minimization problem is replaced with the maximization problem and λ is the regularization parameter. For a large λ, the optimal solution approximately satisfies the equality constraint $\sum_{c}\pi_{c}r\left(i|c\right)=1/n$.

For the above problem we use the majorization-minimization (MM) algorithm [49]. Let $a_{cij},c\in \left[C\right],i,j\in \left[n\right]$ and $b_{ci}$ be the auxiliary positive variables satisfying $a_{cij}=a_{cji},\sum_{c}a_{cij}=1$ and $\sum_{c}b_{cj}=1$. Then, the objective function is bounded below by
≥∑ijSijlog∑cπcr(i|c)r(j|c)+λn∑ilog∑cπcr(i|c)≥∑c,i,jSijacijlogπcr(i|c)r(j|c)acij+λn∑c,ibcilogπcr(i|c)bci.
For fixed $\pi_{c}$ and r(i|c), the optimal $a_{cij}$ and $b_{ci}$ are explicitly obtained. The optimal solutions of $\pi_{c}$ and r(i|c) for a given $a_{cij}$ and $b_{ci}$ are also explicitly obtained. As a result, we obtain the following algorithm to compute the parameters in the Bernoulli mixture model from the similarity matrix *S*. Algorithm 1 is referred to as the SM-to-BM algorithm.

The convergence of the SM-to-BM algorithm is guaranteed from the general argument of the MM algorithm [49].

Note that the SM-to-BM algorithm yields an approximate BMM model, even if the similarity matrix *S* is not completely positive such as the Hub Promoted. However, the approximation accuracy is thought to be not necessarily high, since not-CPSK such as the Hub Promoted does not directly correspond to the exchangeable mixture model (10).

**Algorithm 1:** SM-to-BM algorithm.**Input:** Similarity matrix S=(Sii′)i,i′∈[n] and the number of classes *C*.**Step 1.** Initial values of auxiliary variables acii′ and bci are defined.**Step 2.** Repeat (i) and (ii) until the solution converges to a point: **(i)** For given acii′ and bci:
πc⟵∑i,i′Sii′acii′+λn∑ibci∑c,i,i′Sii′acii′+λn∑c,ibci,r(i|c)⟵2∑i′Sii′acii′+λnbci2∑i,i′Sii′acii′+λn∑ibci**(ii)** For given ric and πc:
acii′⟵ricri′cπc∑cricri′cπc,bci⟵ricπc∑cricπc.**Step 3.** Terminate the algorithm with the output: “The similarity matrix *S* is approximately obtained from the Bernoulli mixture model with πc and the auxiliary variable r(c|i)=n·r(i|c)π(c).”

## 6. Numerical Experiments of Reciprocal Recommendation

We conducted numerical experiments to ensure the effectiveness of the BMMs for the reciprocal recommendation. We also investigated how well the SM-to-BM algorithm works for the recommendation. In numerical experiments, we compare the prediction accuracy for the recommendation problems.

Suppose that there exist two groups, $X=\{x_{1},\ldots ,x_{n}\}$ and $Y=\{y_{1},\ldots ,y_{m}\}$. Expressions of interest between these two groups are observed and they are expressed as directed edges. Hence, the observation is summarized as the bipartite graph with directed edges between *X* and *Y*. If there exists two directed edges (x,y) and (y,x) between x∈X and y∈Y, the pair is a preferable match in the graph. The task is to recommend a subset of *Y* to each element in *X* and vice versa based on the observation. The purpose is to provide potentially preferable matches as much as possible.

There are several criteria used to measure the prediction accuracy. Here, we use the mean average precision (MAP), because the MAP is a typical metric for evaluating the performance of recommender systems; see [5,50,51,52,53,54,55,56] and references therein for more details.

Let us explain the MAP according to [50]. The recommendation to the element *x* is provided as the ordered set of *Y*, i.e., $y_{\left(1\right)},y_{\left(2\right)},\ldots ,y_{\left(m\right)}$, meaning that the preferable match between *x* and $y_{\left(1\right)}$ is regarded to be most likely to occur compared to $y_{\left(2\right)},\ldots ,y_{\left(m\right)}$. Suppose that for each x∈X, the preferable matches with elements in the subset ${\hat{Y}}_{x}\subset Y$ are observed in the test dataset. Let us define $z_{i}$ as $z_{i}=1$ if $y_{\left(i\right)}$ is included in ${\hat{Y}}_{x}$ and otherwise $z_{i}=0$. The precision at the position *k* is defined as $P@k=\frac{1}{k}\sum_{i=1}^{k}z_{i}$. The average precision $\nu_{x}$ is then given as the average of P@k with the weight $z_{k}$, i.e.,
$$\begin{array}{c} \nu_{x}=\frac{\sum_{k=1}^{m}z_{k}P@k}{\sum_{k=1}^{m}z_{k}}. \end{array}$$

Note that $\nu_{x}$ is well defined unless $\sum_{i=1}^{m}z_{i}$ is zero. For example, we have $\nu_{x}=1$ for ${\hat{Y}}_{x}=\left\{y_{\left(1\right)},\ldots ,y_{\left(m^{\prime }\right)}\right\}$ with $m^{\prime }\leq m$, and $\nu_{x}=1-\frac{m^{\prime }}{m-m^{\prime }}\sum_{k=m^{\prime }+1}^{m}\frac{1}{k}$ for ${\hat{Y}}_{x}=\left\{y_{\left(m^{\prime }+1\right)},\ldots ,y_{\left(m\right)}\right\}$ with $0\leq m^{\prime }<m$. In the latter case, $\nu_{x}=1/m$ for $m^{\prime }=m-1$, and $\nu_{x}=\frac{1}{m}+\frac{1}{2(m-1)}$ for $m^{\prime }=m-2$. The MAP is defined as the mean value of $\nu_{x}$ over x∈X. The high MAP value implies that the ordered set over *Y* generated by the recommender system is accurate on average. We use the normalized MAP that is the ratio of the above MAP and the expected MAP for the random recommendation. The normalized MAP is greater than one when the prediction accuracy of the recommendation is higher than that of the random recommendation.

The normalized discounted cumulative gain (NDCG) [5,50,57] is another popular measure in the literature of information retrieval. However, the computation of the NDCG requires the true ranking over the node. Hence, the NDCG is not available for the real-world data in our problem setup.

### 6.1. Gaussian Mixture Models

The graph is randomly generated based on the attributes defined on each node. The size of *X* and *Y* is 1000. Suppose that $x_{i}\in X$ has the profile vector $x_{i}^{\prime }\in R^{100}$ and the preference vector $x_{i}^{\prime \prime }\in R^{100}$. Thus, the attribute vector of $x_{i}$ is given by $\left(x_{i}^{\prime },x_{i}^{\prime \prime }\right)\in R^{200}$. Likewise, the attribute vector $\left(y_{j}^{\prime },y_{j}^{\prime \prime }\right)\in R^{200}$ of $y_{j}\in Y$ consists of the profile vector $y_{j}^{\prime }\in R^{100}$ and the preference vector $y_{j}^{\prime \prime }\in R^{100}$. For each $x_{i}\in X$, 100 elements in *Y*, for example, $y_{k_{1}},\ldots ,y_{k_{100}}$ are randomly sampled. Then, the Euclidean distance between the preference vector $x_{i}^{\prime \prime }$ of $x_{i}$ and the profile vector $y_{k_{j}}^{\prime }$ of $y_{k_{j}}$, i.e., $∥x_{i}^{\prime \prime }-y_{k_{j}}^{\prime }∥$ is calculated for each $y_{k_{j}}$. Then, the 10 closest $y_{k_{j}}$ from $x_{i}$ in terms of the above distance are chosen and directed edges from $x_{i}$ to the 10 chosen nodes in *Y* are added. In the same way, the edges from *Y* to *X* are generated and added to the graph. The training data is obtained as a random bipartite graph. Repeating the same procedure again with a different random seed, we obtain another random graph as a test data.

The above setup imitates practical recommendation problems. Usually, a profile vector is observed for each user. However, the preference vector is not directly observed, while the preference of each user can be inferred via the observed edges.

In our experiments, the profile vectors and preference vectors are identically and independently distributed from the Gaussian mixture distribution with two components, i.e.,
$$\begin{array}{c} x_{i}^{\prime },x_{i}^{\prime \prime },y_{j}^{\prime },y_{j}^{\prime \prime }{∼}_{i.i.d.}\frac{1}{2}N_{100}\left(0,I_{100}\right)+\frac{1}{2}N_{100}\left(1,I_{100}\right), \end{array}$$
meaning that each profile or preference vector is generated from $N_{100}\left(0,I_{100}\right)$ or $N_{100}\left(1,I_{100}\right)$ with probability 1/2. Hence, each node in *X* is roughly categorized into one of two classes, i.e., 0 or 1, that is the mean vector of the preference, $x_{i}^{\prime \prime }$. When the class of $x_{i}$ is 0 (resp. 1), the edge from $x_{i}$ is highly likely to be connected to $y_{j}$ having the profile vector generated from $N_{100}\left(0,I_{100}\right)$ (resp. $N_{100}\left(1,I_{100}\right)$). Therefore, the distribution of edges from *X* to *Y* will be well approximated by the Bernoulli mixture model with C=2. Figure 1 depicts the relationship between the distribution of attributes and edges from *X* to *Y*. The same argument holds for the distribution of edges from *Y* to *X*.

In this simulation, we have focused on the recommendation using similarity measures based on the graph structure. The recommendation to each node of the graph was determined by (Equation 1), where the similarity measures in Table 1 or the one determined from the Bernoulli mixture model (10) were employed. Table 2 shows the averaged MAP scores with the median absolute deviation (MAD) over 10 repetitions with different random seeds. In our experiments, the recommendation based on the BMMs with the appropriate number of components outperformed the other methods. However, the BMMs with a high number of components showed low prediction accuracy.

Below, we show the edge prediction based on the SM-to-BM algorithm in Section 5. The results are shown in Table 3. The number of components in the Bernoulli mixture model was set to C=2 or C=5. Given the similarity matrix *S*, the SM-to-BM algorithm yielded the parameter $\pi_{c}$ and r(i|c). Next, edges were predicted through the formula (10) using $\pi_{c},r\left(i|c\right)$ and $r\left(i\right)=\sum_{c}\pi_{c}r\left(i|c\right)$. The averaged MAP scores of this procedure are reported in the column of “itr:0”. We also examined the edge prediction by the BMMs with the parameter updated from the one obtained by the SM-to-BM algorithm, where the update formula is given by (Equation 7) and (Equation 8). The “itr:10” (resp. “itr:100”) column shows the MAP scores of the edge prediction using 10 times (resp. 100 times) updated parameter. In addition, the “BerMix” shows the MAP score of the BMMs with the updated parameter from the randomly initialized parameter.

In our experiments, we found that the SM-to-BM algorithm applied to commonly used similarity measures improved the accuracy of the recommendation. The MAP score for the “itr:0” method achieved a higher accuracy than the original similarity-based methods. The updated parameter from “itr:0”, however, did not improve the MAP score significantly. The results of “itr:10” and “itr:100” for similarity measures were almost the same when the model was the Bernoulli mixture model with C=2. This is because the EM algorithm with 10 iterations achieved the stationary point of this model in our experiments. We confirmed that there was yet a gap between the likelihood of the parameter computed by the SM-to-BM algorithm and the maximum likelihood. However, the numerical results indicate that the SM-to-BM algorithm provides a good parameter for the recommendation in the sense of the MAP score.

### 6.2. Real-World Data

We show the results for real-world data. The data was provided from an online dating site. The set *X* (resp. *Y*) consists of n=15,925 males and m=16,659 females. The data were gathered from 3 January 2016 to 5 June 2017. We used 130,8126 messages from 3 January 2016 to 31 October 2016 as the training data. Test data consists of 177,450 messages from 1 November 2016 to 5 June 2017 [55]. The proportion of edges in the test set to all data set is approximately 0.12.

In the numerical experiments, half of the users were randomly sampled from each group, and the corresponding subgraph with the training edges were defined as the training graph. On the other hand, the graph with the same nodes in the training graph and the edges in the test edges were used as the test graph. Based on the training graph, the recommendation was provided and was evaluated on the test graph. The same procedure was repeated over 20 times and the averaged MAP scores for each similarity measure were reported in Table 4. In the table, the MAP score of the recommendation for *X* and *Y* were separately reported. So far, we have defined the similarity measure based on out-edges from each node of the directed bipartite graph, referred to as “Interest”. On the other hand, the similarity measure defined by in-edges is referred to as “Attract”. For the BMMs, “Attract” means that the model of each component is computed under the assumption that each in-edge is independently generated, i.e., the probability of ${\left(a_{ij}\right)}_{i\in [n]}$ is given by $\prod_{i}\alpha_{ic}^{a_{ij}}{\left(1-\alpha_{ic}\right)}^{1-a_{ij}}$ when the class of $y_{j}\in Y$ is *c*. In the real-world datasets, the SM-to-BM algorithm was not used, because the dataset was too large to compute the corresponding BMMs from similarity matrices.

As shown in the numerical results, the recommendation based on the BMMs outperformed the other methods. Some similarity measures such as the Common Neighbors or Adamic-Adar coefficient showed relatively good results. On the other hand, the Hub Promoted measure, that is not the CPSK, showed the lowest prediction accuracy. As well as the result for synthetic data, the BMMs with two to five components produced high prediction accuracy. Even for medium to large datasets, we found that the Bernoulli mixture model with about five components worked well. We expect that the validation technique is available to determine the appropriate size of components. Also, the similarity with “Interest” or “Attract” can be determined from the validation dataset.

## 7. Discussions and Concluding Remarks

In this paper, we considered the relationship between the similarity-based recommendation methods and statistical models. We showed that the BMMs are closely related to the recommendation using completely positive similarity measures. More concretely, both the BMM-based method and completely positive similarity measures share exchangeable mixture models as the statistical model of the edge distribution. Once this was established, we proposed the recommendation methods using the EM algorithm to BMMs to improve similarity-based methods.

Moreover, we proposed the SM-to-BM algorithm that transforms a similarity matrix to parameters of the Bernoulli mixture model. The main purpose of the SM-to-BM algorithm is to find a statistical model corresponding to a given similarity matrix. This transformation provides a statistical interpretation for similarity-based methods. For example, the conditional probability r(c|i) is obtained from the SM-to-BM algorithm. This probability is useful to categorize nodes, i.e., users, into some classes according to the tendency of their preferences once a similarity matrix is obtained. The SM-to-BM algorithm is available as a supplementary tool for similarity-based methods.

We conducted numerical experiments using synthetic and real-world data. We numerically verified the efficiency of the BMM-based method in comparison to similarity-based methods. For the synthetic data, the BMM-based method was compared with the recommendation using the statistical model obtained by the SM-to-BM algorithm. We found that the BMM-based method and the SM-to-BM method provide a comparable accuracy for the reciprocal recommendation. Since the synthetic data is well approximated by the BMM with C=2, the SM-to-BM algorithm was thought to reduce the noise in the similarity matrices. In the real-world data, the SM-to-BM algorithm was not examined, since our algorithm using the MM method was computationally demanding for a large dataset. On the other hand, we observed that the BMM-based EM algorithm was scalable for a large dataset. A future work includes the development of computationally efficient SM-to-BM algorithms.

It is straightforward to show that the stochastic block models (SBMs) [6] are also closely related to the recommendation with completely positive similarity measures. In our preliminary experiments, however, we found that the recommendation system based on the SBMs did not show a high prediction accuracy in comparison to other methods. We expect that detailed theoretical analysis of the relation between the similarity measure and statistical models is an interesting research topic that can be used to better understand the meaning of the commonly used similarity measures.

## Figures and Tables

**Figure 1 entropy-21-00702-f001:**
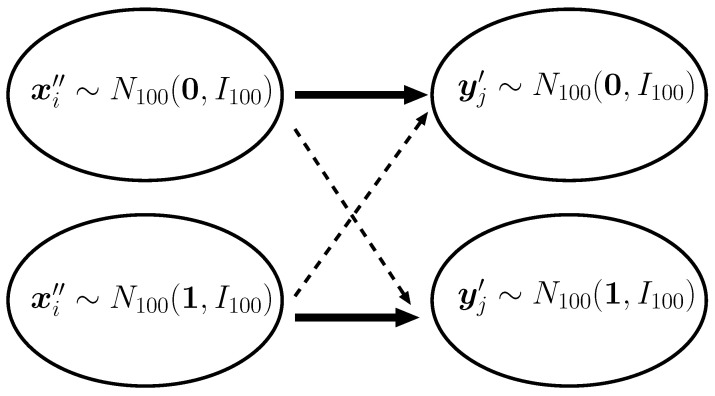
Edges from *X* to *Y*. The bold edges mean that there are many edges between the connected groups. The broken edges mean that there are few edges between the connected groups.

**Table 1 entropy-21-00702-t001:** Definition of similarity measures $\mathrm{sim}(i,i^{\prime })$ between the nodes $x_{i}$ and $x_{i^{\prime }}$. The right column shows whether the similarity measure is a completely positive similarity kernel (CPSK); see Section 4.

Similarity	Definition/Condition of $S=(\mathrm{sim}\left(i,i^{\prime }\right))$	CPSK
Common neighbors [27]	$|s_{i}\cap s_{i^{\prime }}|$	*√*
Parameter-dependent [28]	$|s_{i}\cap s_{i^{\prime }}\left|/(\right|s_{i}\left|\right|s_{i^{\prime }}{\left|\right)}^{\lambda },\;\lambda \geq 0$	*√*
Jaccard coefficient [29]	$|s_{i}\cap s_{i^{\prime }}\left|/\right|s_{i}\cup s_{i^{\prime }}|$	*√*
Sørensen index [30]	$|s_{i}\cap s_{i^{\prime }}\left|/(\right|s_{i}\left|+\right|s_{i^{\prime }}\left|\right)$	*√*
Hub depressed [31]	$|s_{i}\cap s_{i^{\prime }}\left|/\max\{\right|s_{i}\left|,\right|s_{i^{\prime }}\left|\right\}$	*√*
Hub promoted [32]	$|s_{i}\cap s_{i^{\prime }}\left|/\min\{\right|s_{i}\left|,\right|s_{i^{\prime }}\left|\right\}$	×
SimRank [33]	$S=cP^{T}SP+D$, c∈(0,1)	*√*
Adamic-adar coefficient [34]	$\sum_{k\in s_{i}\cap s_{i^{\prime }}}1/\log\left|{\bar{s}}_{k}\right|$	*√*
Resource allocation [31]	$\sum_{k\in s_{i}\cap s_{i^{\prime }}}1/\left|{\bar{s}}_{k}\right|$	*√*
Content-based similarity [17]	$\frac{1}{A}\sum_{a\in [A]}\sum_{b\in V_{a}}1\left[{\left(x_{i}\right)}_{a}=b,{\left(x_{i^{\prime }}\right)}_{a}=b\right]$	*√*

**Table 2 entropy-21-00702-t002:** Mean average precision (MAP) values of similarity-based methods under synthetic data. The bold face indicates the top two MAP scores.

Similarity	MAP (MAD)
Common Neighbors	1.889 (±0.413)
Cosine	1.907 (±0.431)
Jaccard Coefficient	2.115 (±0.421)
Sørensen Index	2.021 (±0.369)
Hub Depressed	2.231 (±0.376)
Hub Promoted	2.053 (±0.301)
SimRank	2.853 (±0.610)
Adamic-Adar coefficient	2.188 (±0.587)
Resource Allocation	1.950 (±0.516)
Bernoulli Mixture (C=2)	**5.599** (±1.811)
Bernoulli Mixture (C=5)	**4.552** (±1.766)
Bernoulli Mixture (C=10)	2.821 (±1.164)
Bernoulli Mixture (C=50)	1.382 (±0.449)
Bernoulli Mixture (C=100)	1.535 (±0.555)

**Table 3 entropy-21-00702-t003:** MAP values of updated Bernoulli mixture models with the SM-to-BM algorithm under synthetic data. The results of Bernoulli mixture models with C=2 and C=5 are reported. The bold face indicates the top two MAP scores in each column.

	**SM-to-BM:** C=2		
**Similarity**	**itr:0**	**itr:10**	**itr:100**
Common Neighbors	**5.646** (±1.041)	**5.572** (±1.041)	**5.572** (±1.041)
Cosine	4.725 (±1.119)	4.549 (±1.119)	4.549 (±1.119)
Jaccard Coefficient	**5.421** (±0.643)	5.373 (±0.643)	5.373 (±0.643)
Sørensen Index	5.013 (±2.223)	4.964 (±2.223)	4.964 (±2.223)
Hub Depressed	5.417 (±1.756)	5.262 (±1.756)	5.262 (±1.756)
Hub Promoted	5.120 (±0.563)	5.165 (±0.563)	5.165 (±0.563)
SimRank	3.848 (±1.630)	4.377 (±1.279)	4.379 (±1.264)
Adamic-Adar coefficient	4.348 (±1.170)	4.404 (±1.170)	4.404 (±1.170)
Resource Allocation	4.435 (±0.552)	4.385 (±0.552)	4.385 (±0.552)
BerMix. (Random ini.)	1.297 (±0.446)	**5.718** (±2.013)	**5.911** (±2.087)
	**SM-to-BM:** C=5		
**Similarity**	**itr:0**	**itr:10**	**itr:100**
Common Neighbors:	5.059 (±0.939)	**5.557** (±0.939)	**5.137** (±0.939)
Cosine	4.442 (±0.901)	4.070 (±0.901)	3.948 (±0.901)
Jaccard Coefficient	5.167 (±1.745)	4.765 (±1.745)	4.792 (±1.745)
Sørensen Index	**5.675** (±0.773)	**5.294** (±0.773)	**5.189** (±0.773)
Hub Depressed	**5.408** (±1.807)	4.668 (±1.807)	4.391 (±1.807)
Hub Promoted	5.078 (±0.702)	4.815 (±0.702)	5.008 (±0.702)
SimRank	4.121 (±1.274)	3.592 (±1.150)	3.615 (±1.447)
Adamic-Adar coefficient	5.284 (±1.166)	4.909 (±1.166)	5.084 (±1.166)
Resource Allocation	4.884 (±0.751)	4.499 (±0.751)	4.263 (±0.751)
BerMix. (Random ini.)	1.080 (±0.446)	3.705 (±1.925)	4.810 (±1.268)

**Table 4 entropy-21-00702-t004:** MAP scores for real-world data. The bold face indicates the top two MAP scores in each column.

Similarity	Recomm. of *Y* to *X*	Recomm. of *X* to *Y*
	MAP (MAD):	MAP (MAD)
Common Neighbors:Interest	6.267 (±0.806)	2.893 (±0.343)
Common Neighbors:Attract	2.053 (±0.276)	8.813 (±0.757)
Cosine:Interest	3.496 (±0.324)	3.699 (±0.309)
Cosine:Attract	2.746 (±0.276)	6.108(±0.546)
Jaccard Coefficient:Interest	4.098 (±0.373)	4.066 (±0.294)
Jaccard Coefficient:Attract	3.288 (±0.362)	7.777 (±0.724)
Sørensen Index:Interest	4.205 (±0.363)	3.996 (±0.293)
Sørensen Index:Attract	3.205 (±0.319)	7.910 (±0.620)
Hub Depressed:Interest	4.370 (±0.369)	4.106 (±0.291)
Hub Depressed:Attract	3.366 (±0.379)	8.364 (±0.613)
Hub Promoted:Interest	1.959 (±0.300)	2.691 (±0.334)
Hub Promoted:Attract	1.662 (±0.262)	2.641 (±0.263)
SimRank:Interest	2.079 (±0.164)	6.336 (±0.423)
SimRank:Attract	5.100 (±0.775)	3.158 (±0.193)
Adamic-Adar coefficient:Interest	6.216 (±0.701)	2.970 (±0.308)
Adamic-Adar coefficient:Attract	2.209 (±0.267)	8.300 (±0.632)
Resource Allocation:Interest	5.521 (±0.679)	3.557 (±0.262)
Resource Allocation:Attract	2.713 (±0.298)	6.875 (±0.660)
Bernoulli Mixture (C=2):Interest	4.578 (±0.734)	**15.061** (±4.106)
Bernoulli Mixture (C=2):Attract	**10.625** (±2.054)	12.825 (±2.323)
Bernoulli Mixture (C=5):Interest	5.055 (±0.813)	**17.394** (±3.271)
Bernoulli Mixture (C=5):Attract	10.362 (±1.981)	10.348(±2.514)
Bernoulli Mixture (C=10):Interest	4.263 (±0.772)	15.013 (±4.042)
Bernoulli Mixture (C=10):Attract	**10.451** (±1.133)	8.786 (±1.195)
Bernoulli Mixture (C=50):Interest	5.664 (±1.873)	14.288 (±6.409)
Bernoulli Mixture (C=50):Attract	10.029 (±3.933)	8.436 (±3.929)
Bernoulli Mixture (C=100):Interest	2.910 (±0.436)	8.525 (±1.199)
Bernoulli Mixture (C=100):Attract	5.980 (±1.612)	5.119 (±0.464)

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
