# Peer review of "Model Description of Similarity-Based Recommendation Systems"

_entropy, 2019, doi:10.3390/e21070702_

Round 1
Reviewer 1 Report
Overall paper is nicely drafted and presented. Experiments are very detail. I feel that the authors should rephrase the introduction section to mention the work be done and highlight the achievements in terms of the results.
Author Response
We deeply appreciate the reviewer's positive comments. In the revised manuscript, we substantially revised the wordings and phrases.Reviewer 2 Report
The authors investigate the possibility to feed statistical network models with node similarity values obtained by one of the usual similarity metrics for network nodes. Experiments on synthetic and real datasets show that the proposed ideas work well in practice. The manuscript is theoretically sound and the presented results are relevant to other researchers working in network science and information filtering in particular. The manuscript's level of exposition is good but lacks clarity in many places --- I try to point out the most significant ones in the list of comments below. By reflecting on my comments, the manuscript will become more accessible to the readers and suitable for publication.
1) EM in the abstract would be better to be spelled out.
2) Introduction is somewhat confusing when it comes to what is the target object of your study. You speak first about social networks, and provide the network of job seekers and companies as an example though it is not really a typical social network (which, as you say, consists of users and relations between them). Then you continue with discussing other bipartite networks, and then there is again a change at the end of Introduction when you introduce notation for directed monoparite networks, whereas the reader has no idea why they are relevant (and, related, why no notation for bipartite networks is introduced). If you consider directed monopartite networks in your mathematical analysis, then tailor Introduction and the examples therein to that.
3) Line 40: "is derived" -> "can be derived"? Also, does this hold for all similarity-based methods? You prove that they are "completely positive" only for some of them...
4) Line 44: What is "probabilistic variant" of a matrix? (Here, you say "probabilistic variant of completely positive matrices".)
5) What are the node attributes mentioned in line 59 (mathematical form, what is their role/purpose, where do they come from)?
6) If you indeed consider directed networks, then one should distinguish between downstream and upstream neighbors of node i. Why do you automatically assume that the common neighbors similarity (as well as most other metrics Table 1) is determined by s_i and s_j, and not by s_i' and s_j'? In Section 6.2, I see that you actually consider both - you can mention that in the initial definitions.
7) Most works on link prediction use undirected networks. How does your formalism apply to undirected networks? I have the same question with respect to reciprocal recommendation/link prediction which, again, is a subset of recommendation/link prediction in directed networks. How do your results apply to directed recommendation/link prediction?
8) When you speak about the independence on the graph structure (line 93), there are first some grammar issues which make the sentence difficult to follow, and second the attribute vectors typically influence/are determined from the network structure, so it's not really true that
9) When introducing BMMs (Section 3), you first say that pi_c is the probability that the edge is for a given class but then speak about nodes being from a class actually. So no edges, just nodes, right?
10) Line 110: A reference would be useful here. The Expectation-Maximization (EM) algorithm in the next line deserves a reference too. In general, it could be good to add references to the applications of BMMs in networks.
11) What is the reason for enforcing the first of the three constraints in the optimization problem on page 11 only approximately?
12) The choice of the mean average prediction as the only link prediction evaluation metric is very unfortunate. Personally, I have not heard of this metric before and the reference is to a technical report by an institute in Japan which cannot be accessed (at least Google Scholar does not get me to the pdf) and it has not been greatly cited (9 citations in 14 years). (From the MAP's definition, it seems unusual that it is quadratic in terms of z; but I do not really follow the definition and rationale behind v_x, so maybe it is alright.) Why not to use some common evaluation metrics (precision, recall, F1, area under the precision-recall curve, etc.) instead? Different evaluation metrics put emphasis on different aspects of evaluation, so having results using more than one of them is in any case beneficial.
13) The explanation of the synthetic data in Section 6.1 is too brief without clearly explaining what is the point of introducing profile and preference vectors, what is their difference from attribute vectors introduced before, etc.
14) You have not built any classes in your synthetic data, right? Why does then the Bernoulli Mixture model performs best with only two classes which is much less than both the number of nodes (1000) as well as the attribute/profile/preference space dimension (100)?
15) What is the proportion between the training and test links in Section 6.2? The discussion of results here is very very brief. It would be good to discuss/clarify what is both practical and methodological difference between the BMNs and the SM-to-BM method. (I know that both are explained before but that's in separation, so it could be helpful to stress the similarities and differences between the two approaches.)
16) I am puzzled as to what is the significance of assessing/proving the complete positivity of individual metrics. From Section 3, I would think that you can do the SM-to-BM only for a complete positive metric but then in Table 3 you have results also for Hub Promoted index, so I do not really understand where the complete positivity enters and why it is important.
17) It is a pity that many equations, including some important ones, are not numbered.
18) While the overall level of writing is good, English still can be improved at many places (such as the singular/plural conflict (and articles) in the heading in line 196).
Author Response
Dear Reviewers,
We sincerely appreciate the valuable comments of the editor and the reviewers on our manuscript. We revised our manuscript following your suggestions, which have contributed to substantial improvements to the earlier version of the work.
